# Assessment of Maternal Exposure to Mycotoxins During Pregnancy Through Biomarkers in Fetal and Neonatal Tissues

**DOI:** 10.3390/toxins17100518

**Published:** 2025-10-21

**Authors:** João Thiago Aragão Fermiano, Sher Ali, Sana Ullah, Vanessa Theodoro Rezende, Roice Eliana Rosim, Fernando Gustavo Tonin, Walusa Assad Gonçalves Ferri, Alessandra Cristina Marcolin, Leandra Naira Zambelli Ramalho, Carlos Augusto Fernandes de Oliveira, Fernando Silva Ramalho

**Affiliations:** 1Department of Pathology, Ribeirão Preto Medical School, University of São Paulo, Ribeirão Preto 14049-900, SP, Brazil; joaothiago@usp.br (J.T.A.F.); lramalho@fmrp.usp.br (L.N.Z.R.); 2Department of Food Engineering, School of Animal Science and Food Engineering, University of São Paulo, Pirassununga 13635-900, SP, Brazil; alisher@usp.br (S.A.); sanaullah@usp.br (S.U.); roice@usp.br (R.E.R.); 3Faculty of Veterinary and Animal Science, University of São Paulo, Pirassununga 13635-900, SP, Brazil; vanessatrezende@usp.br; 4Department of Biosystems Engineering, School of Animal Science and Food Engineering, University of São Paulo, Pirassununga 13635-900, SP, Brazil; fgtonin@usp.br; 5Department of Pediatrics, Ribeirão Preto Medical School, University of São Paulo, Ribeirão Preto 14049-900, SP, Brazil; walusa@fmrp.usp.br; 6Department of Obstetrics and Gynecology, Ribeirão Preto Medical School, University of São Paulo, Ribeirão Preto 14049-900, SP, Brazil; amarcolin@fmrp.usp.br

**Keywords:** mycotoxins, biomarkers, prenatal exposure, developmental toxicity, congenital defects, UPLC-MS/MS

## Abstract

This study aimed to conduct a first evaluation of maternal exposure to mycotoxins during pregnancy in Brazil through quantification of biomarkers in liver and serum samples from stillborn and neonates autopsied in the Clinical Hospital of Ribeirão Preto, state of São Paulo. Liver tissue (*n* = 43) and serum (*n* = 38) samples were collected from 43 patients and analyzed for biomarkers of aflatoxins (AFs), ochratoxin A (OTA), fumonisins (FBs), zearalenone (ZEN), deoxynivalenol (DON), T-2 and HT-2 toxins by ultra-performance liquid chromatography coupled to tandem mass spectrometry. In total, 9 samples of liver (20.9%) and 33 serum (86.8%) had quantifiable levels of mycotoxins. In liver samples, aflatoxin M_1_ (AFM_1_) was quantified in two samples (median level: 0.38 ng/g), while four samples had OTA residues (median: 0.31 ng/g) and one contained ZEN (3.6 ng/g). Compared with liver tissue, serum samples had higher occurrence rates of mycotoxins, particularly AFM_1_, OTA and ZEN. Nineteen serum samples (50%) contained 2–4 types of mycotoxins, indicating an effective transplacental transfer of major mycotoxins during pregnancy. Median levels of AFM_1_, OTA, FB_1_, ZEN, DON, T-2 and HT-2 toxins in serum samples were 0.48, 3.39, 30.6, 10.53, 5.71, 2.85 and 10.84 ng/mL, respectively. The most frequent cause of death was extreme prematurity (33% of cases), followed by preterm premature rupture of membranes (16% of cases) and morphological abnormalities (42% of cases). Results of this trial suggest potential associations between dietary mycotoxins and congenital anomalies. Further research should clarify the transplacental transfer of mycotoxins and their association with toxic effects during human prenatal development.

## 1. Introduction

The global food safety systems face a persistent contamination threat by mycotoxin-producing fungi, especially from species belonging to the genera *Aspergillus*, *Penicillium*, and *Fusarium* [1,2]. These fungi can proliferate throughout the food production chain, from cultivation to storage. Factors such as substrate availability, high humidity, insect infestation, improper processing, and poor storage conditions significantly contribute to fungal growth and production of diverse mycotoxins in foods worldwide, including in Brazil [1,2,3]. The most frequently detected mycotoxins in plant-based foods include the aflatoxins (AFs) of the B and G types (e.g., AFB_1_, AFB_2_, AFG_1_, AFG_2_), mainly produced by *A. flavus* and *A. parasiticus*; ochratoxin A (OTA), primarily produced by members of *Aspergillus* section Circumdati and some *Penicillium* species; fumonisins (F) B_1_, B_2_ and B_3_, synthesized mainly by *F. verticillioides*; zearalenone (ZEN) and deoxynivalenol (DON), largely produced by *F. graminearum*; and type A trichothecenes (e.g., T-2 and HT-2 toxins) produced by other *Fusarium* species such as *F. sporotrichioides* and *F. langsethiae* [1,2,4,5,6,7]. The International Agency for Research on Cancer (IARC) has classified AFB_1_ in Group 1 (carcinogen to humans), FB_1_ and OTA in Group 2B (possible carcinogens to humans), and ZEN, DON, T-2 and HT-2 toxins in Group 3 (unclassifiable as carcinogens to humans) [8]. Studies have consistently described human exposure to such toxicants across diverse food categories, especially corn, wheat, rice, bean and peanuts [1,2,9,10]. In addition, livestock fed mycotoxin-contaminated feed can transfer their residues into animal-derived foods, especially in milk and derived products [1,2]. In Brazil, maximum levels (MLs) for mycotoxins in foods have been recently reviewed by the National Health Regulatory Agency (ANVISA), setting up ML for AF (1–20 μg/kg), OTA (2–20 μg/kg), FBs (200–4000 μg/kg), ZEN (20–600 μg/kg) and DON (200–2000 μg/kg) in several food categories [11]. However, despite these regulations, the frequent occurrence of mycotoxins in foods continues to pose significant health risks in Brazil [1,2,12].

The dietary exposure to mycotoxin poses a significant health risk, particularly during the prenatal period. This critical window, spanning from conception to birth, encompasses key stages of organogenesis, tissue differentiation, and physiological maturation. Each trimester presents specific biological vulnerabilities, with the first trimester being especially sensitive to environmental toxicants that can impair prenatal development [13]. During this period, rapid growth and maturation of neural, cardiovascular, immune, skeletal, and endocrine systems make the fetus highly susceptible to even low exposure levels to mycotoxins. In placental mammals, clinical evaluations indicate that AFs may adversely affect not only the fetus but also the pregnant individual and the maternal–fetal interface [14]. Depending on the type and dose, mycotoxins can exert various developmental toxicities, including intrauterine growth restriction and morphological abnormalities, sometimes resulting in fetal death [15]. Mycotoxins are generally capable of crossing the placental barrier, thereby exposing the fetus to harmful insults during critical developmental stages [16,17]. Consequently, maternal diet is considered a primary route of fetal exposure to mycotoxins. This is particularly critical during early development, as fetal detoxification systems are immature and unable to effectively eliminate mycotoxins [16,17]. However, the transfer rates of mycotoxins from the maternal blood through the fetus during human pregnancy are not known. Additionally, there are no available studies on the maternal exposure to dietary mycotoxins during pregnancy in Brazil.

Human exposure assessments to dietary mycotoxins have been performed based on consumption patterns and the occurrence levels of these toxicants in food products [17]. However, limitations of this approach may include heterogeneous distribution of mycotoxins in food products and limited accuracy of dietary recall questionnaires [18]. In recent years, biomarkers have been successfully used as suitable alternatives for a more accurate evaluation of individual exposure to mycotoxins [18]. Animal and human studies demonstrated that several mycotoxins measured in bio-specimens such as liver and serum correlate with dietary mycotoxin ingestion [18,19]. In addition, after absorption in the gastrointestinal tract, some mycotoxins undergo phase I biotransformation, hence originating metabolites that also serve as biomarkers of exposure to the parent compound. For example, after ingestion, AFB_1_ is biotransformed to its hydroxylated metabolite aflatoxin M_1_ (AFM_1_), among other compounds, in the liver by the cytochrome P450 system [20]. ZEN is primarily metabolized in the liver through phase 1 reductions into α-zearalenol (α-ZEL) and β-zearalenol (β-ZEL), which can also remain quantifiable in tissues and body fluids, indicating ZEN exposure [18,19]. DON is metabolized by the microbiota in the upper gastrointestinal tract into de-epoxy-deoxynivalenol (DOM-1) [18]. However, ZEN and DON can also originate phase II metabolites such as glucuronide or sulfate conjugates [12]. All these conjugated or non-conjugated metabolites, along with their parent compounds and other unmetabolized mycotoxins such as OTA, FB1, T-2 and HT-2 toxins, have been used as biomarkers to estimate the exposure of pregnant women to dietary mycotoxins, and to assess fetotoxicity or birth outcomes [9,10]. Several studies have described associations between maternal exposure to some mycotoxins and adverse birth outcomes, often through indirect assessments of biomarkers in maternal urine or serum [9,10,21,22]. However, there is very little information on the occurrence levels of mycotoxins measured directly in fetal bio-specimens (e.g., liver, serum). This gap limits the knowledge on the carry-over of mycotoxins and potential toxic effects on fetuses during human pregnancy. This study aimed to conduct a first evaluation of maternal exposure to dietary mycotoxins during pregnancy in Brazil through quantification of AFs, OTA, FBs, DON, T-2 and HT-2 toxins, ZEN and its metabolites (α-ZEL and β-ZEL) in liver and serum samples from stillborn and neonates autopsied at a university hospital.

## 2. Results

### 2.1. Stillborns and Neonatal Demographic and Clinical Characteristics

A total of 43 patients were included in this study, comprising stillborns (*n* = 20) and neonates (*n* = 23), all of whom underwent autopsy examinations at the Clinical Hospital of Ribeirão Preto (CHRP), Ribeirão Preto Medical School, University of São Paulo, Brazil. The complete demographic, clinical, and morphological findings for each case are detailed in Table 1, along with the types of mycotoxins quantified in the respective liver tissue and serum samples. The mean and median gestational ages of the stillborns were 28.3 and 28.7 weeks, respectively. The neonates were less than one week old, remained hospitalized in the intensive care unit, and received no oral feeding. Their mean and median postnatal ages were 23.4 and 9 h, respectively.

The most frequent cause of death was recorded as extreme prematurity (gestational age < 28 weeks), accounting for 33% of cases. This was followed by preterm premature rupture of membranes, defined as rupture of the amniochorionic membrane before 37 weeks of gestation, observed in 16% of cases. Morphological abnormalities resulting from intrauterine growth restriction (IUGR) and/or congenital malformations were recorded in 42% (*n* = 18/43) of cases. Except for some cases associated with maternal preeclampsia, HELLP (Hemolysis, Elevated Liver enzymes, and Low Platelet count) syndrome, or sepsis, numerous others demonstrated congenital defects. These congenital anomalies involved severe heart defects (*n* = 3), chromosomal syndromes (Turner and Edwards; *n* = 3), renal abnormalities (hydronephrosis and multicystic dysplastic kidney; *n* = 3), imperfect twinning (*n* = 1), isolated hydrocephalus (*n* = 1), and multiple organ malformations (*n* = 2) (Table 1), including the one described in Figure 1 (case: 1082/2023). This neonate had multiple congenital malformations, including facial dysmorphism, limb deformities, and abnormal pulmonary anatomy. Several critical dysmorphic features were notable and involved a midline facial cleft with bilateral oronasal malformation (Figure 1A), pulmonary hypoplasia with abnormal lobulation and septation (Figure 1B), as well as limb anomalies and likely syndactyly and hypoplastic digits (Figure 1C–E). Samples collected from this neonate had quantifiable levels of OTA in the liver, as well as OTA, DON and T-2 toxin in serum, as summarized in Table 1.

The analysis of serum samples from stillborns and neonates also indicated frequent co-occurrences of mycotoxins, as displayed in Table 1. Nineteen serum samples (50%) contained 2–4 types of mycotoxins, with AFM_1_ + ZEN, OTA + FB_1_ and OTA + DON as the most frequent binary combinations. Four samples contained three types of mycotoxins, and two samples had four types of mycotoxins, one containing AFM_1_ + FB_1_ + ZEN, β-ZEL + HT-2 toxin, and another with AFM_1_ + OTA + ZEN + DON (Table 1).

### 2.2. Occurrence and Levels of Mycotoxins in Liver Tissue and Serum Samples

Table 2 presents the mycotoxins quantified in 43 liver tissue samples and 38 serum samples analyzed in the study. In total, 9 samples of liver and 33 serum samples had quantifiable levels of mycotoxins. In liver samples, AFM_1_ was quantified in two samples (median level: 0.38 ng/g), one of them also containing AFB_1_, AFB_2_ and AFG_1_ at levels ranging from 0.30 to 1.79 ng/g. Notably, four samples had OTA residues (median: 0.31 ng/g), suggesting significant maternal exposure to this mycotoxin. Finally, ZEN and α-ZEL were quantified in two different samples at 3.60 and 2.76 nag/g, respectively. Compared with liver tissue, serum samples had higher occurrence rates of mycotoxins, particularly for AFM_1_, OTA and ZEN, thus highlighting the relevance of serum as a sensitive matrix for biomonitoring neonatal exposure to mycotoxins. AFM_1_ was the only type of AFs quantified in 15 samples of serum (39.5%), at a median level of 0.48 ng/mL. OTA and ZEN were found in 13 samples (34.2%) at median levels of 3.39 and 10.53 ng/mL, respectively. However, a wide range of concentrations was observed, varying from 1.11 to 157.8 ng/mL for OTA, and from 1.32 to 198.9 ng/mL for ZEN. However, ZEN metabolites (α- and β-ZEL) were quantified only in two separate samples. FB_1_, DON, T-2 and HT-2 toxins were quantified in seven (18.4%), nine (23.7%), one (2.6%) and four (10.5%) serum samples, respectively. Although all patients who had quantifiable levels of mycotoxins in the liver (*n* = 7) also exhibited mycotoxins in their serum samples, no correlation (*p* = 0.988) was found between the respective values (Appendix A).

## 3. Discussion

The occurrence levels of the studied mycotoxins in the present work indicate a high maternal exposure to mycotoxin-contaminated foods, and effective transplacental transfer of major mycotoxins during pregnancy. Despite well-established toxicological evidence using animal models on the effects of several mycotoxins during pregnancy, there is a noteworthy lack of studies reporting mycotoxins directly in human fetal or neonatal biospecimens. To the best of our knowledge, the present study is the first to report the simultaneous detection of several mycotoxins in stillborn fetal/neonatal serum and liver tissues alongside clinical pathologies, offering direct evidence of prenatal in utero exposure. It is important to highlight that most of the evaluated mycotoxins exhibit significant teratogenic potential [23,24]. In the present study, the frequency of congenital malformations and/or IUGR was 33% in cases with liver samples containing mycotoxin residues, and 43% in cases with serum samples contaminated with up to four types of mycotoxins. The high frequency of congenital morphological abnormalities documented here can be justified considering that the university hospital (CHRP) is a public tertiary health service, which drains highly complex cases from a population of approximately 2 million people, mostly from low-income Brazilian families.

In our work, it was not possible to establish a close association between the residual levels of mycotoxins found in liver tissue or serum samples and a set of congenital anomalies compatible with specific toxic effects of the evaluated mycotoxins. However, the clinical findings described in this study are in agreement with the available literature regarding the toxic properties of AFs, FB_1_, OTA, ZEN and trichothecenes such as DON, T-2 and HT-2 toxins. AFM_1_ was quantified in 15 serum cases and 2 liver cases (Table 2), indicating that AFs are frequent contaminants in the maternal diets of the studied cases. This is consistent with the high occurrence levels of AFs in foods reported in Brazil [25,26]. Considering the serum data alone reported in the present study, seven cases exhibited malformations and/or IUGR, which are reported outcomes from prenatal exposure to AFs [23]. Human studies from several regions have demonstrated similar exposure and risks. In Poland, a recent study by Gromadzka et al. [27] detected mycotoxin levels in amniotic fluid samples (*n* = 86) collected via abdominal amniocentesis at 15–22 weeks of gestation from pregnant women with a high risk of chromosomal anomalies or genetic fetal defects. The most frequent mycotoxins were AFs, OTA and trichothecenes, detected in over 75% of all samples, and in 73% samples of amniotic fluid from fetuses with genetic defects. In Uganda, a cohort study with 220 mother–infant pairs indicated that maternal AFB_1_-lysine levels (median 5.83 pg/mg albumin) during pregnancy were significantly associated with adverse birth outcomes, including lower birth weight and smaller head circumference [28]. In the AflaCohort study conducted in Nepal involving 1621 mother–infant pairs, maternal AFB_1_-lysine levels (geometric mean 1.37 pg/mg albumin) were significantly associated with increased odds of small-for-gestational-age (SGA) births, though no associations were found for the preterm birth or stunting issues [29]. Likewise, Tesfamariam et al. [9] found AFB_1_-lysine in over 81% of maternal serum samples (median 12.9 pg/mg albumin) in Ethiopia, underscoring the potential for in utero toxicity and impaired fetal organogenesis. Similarly, Shuaib et al. [30] demonstrated that high levels of AFB_1_-lysine (>11.34 pg/mg) in serum samples of Ghanaian pregnant women were associated with a trend of increasing risk for low birth weight. In Nigeria, Abulu et al. [31] reported lower birth weight in jaundiced neonates with high AFB_1_ levels in cord blood samples compared to non-exposed jaundiced neonates. The detection rate of AFB_1_ was higher in the wet (81.8%) than in the dry season (50%). Another Ethiopian study also highlighted widespread maternal exposure to multiple mycotoxins, though no significant associations or adverse birth outcomes were observed [10]. Collectively, these studies corroborate our results and emphasize that maternal AFs exposure remains a persistent concern across diverse populations.

In our work, the quantification of OTA in the liver of patient 1082/2023 and in a fetus affected by IUGR (patient 83/2023) indicates fetal contamination due to OTA’s ability to cross the placental barrier. This observation is consistent with a biomonitoring study reporting OTA in more than 50% of maternal blood samples and in 38.3% of newborns within 12 h of birth in Rural Burkina Faso [21]. In particular, patient 1082/2023 evaluated in the present study exhibited exceptionally high serum levels of OTA (157.8 ng/mL) along with low concentrations of DON (4.17 ng/mL) and HT-2 (13.14 ng/mL), and low level of OTA (0.30 ng/g) in the liver, suggesting significant in utero exposure to different mycotoxins (Table 2). Notably, the same patient (1082/2023) presented bilateral cleft lip and palate, a defect that aligns with teratogenic outcomes observed in animal models. In rodents, rabbits, and hamsters, prenatal OTA exposure led to craniofacial malformations, including cleft lip and palate, micrognathia, and ocular abnormalities, supporting a causal relationship between OTA and orofacial anomalies [32,33,34]. Mechanistically, such defects are linked to disrupted calcium homeostasis affecting ossification processes [34,35,36]. The IUGR patient (83/2023) evaluated here also had OTA in the liver (0.32 ng/g) and AFM_1_ in serum (0.22 ng/mL). In a cohort of pregnant women (*n* = 50) from Connecticut, United States (U.S.), OTA was found to be the most frequent mycotoxin in serum samples (median: 0.09 ng/mL; 46% detection rate) [20]. The authors emphasized the neoplastic risk of OTA (estimated margin of exposure: <10,000) across pregnant women, also suggesting long-term health implications of maternal OTA exposure during pregnancy [37]. High levels of OTA were also identified in serum samples from Czech women (*n* = 100) in the first trimester of pregnancy, considering that OTA crosses the placenta at early gestation rather than in late gestation. The authors observed that 96% of tested maternal serum samples were positive for OTA at concentrations ranging from 0.1 to 0.35 µg/L, with a mean value of 0.15 µg/L [22]. Further epidemiological studies associated OTA exposure with impaired fetal growth in Bangladesh, where OTA was detected in 95% samples of urine from pregnant women, whose concentrations correlated with enhanced odds of low birth weight [38]. Similarly, in Burkina Faso, OTA exposure in 38.3% of newborns was significantly associated with lower birth weight [β −0.11 kg (95% CI: −0.21, 0.00)], lower ponderal index [β −0.62 g/cm^3^ (95% CI: −1.19, −0.05)], as well as marginal reductions in growth during the first six months of life [21]. Such collective evidences corroborate the growth restriction observed in the IUGR patient (83/2023) described in the present study. The clinical histories of patients 83/2023 and 1082/2023 evaluated in our work align with the literature regarding the deleterious effects of OTA on fetal development in animal models, resulting in nephropathies, skeletal deformities, and impaired intrauterine growth [20,39,40]. OTA characteristics, including high lipophilicity, efficient gastrointestinal absorption, strong albumin binding, and limited biotransformation, contribute to OTA’s persistence in the human body [35]. The placenta permits OTA transfer during early gestation, enabling its teratogenic impact by interfering with fetal organogenesis [32]. OTA’s reproductive toxicity is evident in ovarian dysfunction. Studies also demonstrated that OTA exposure led to follicular atresia, mitochondrial damage, meiotic failure, and disrupted epigenetic regulation, culminating in impaired oocyte maturation and reduced fertility [41]. Additionally, OTA-induced clastogenicity in the oogenesis stage is linked to reproductive toxicity, including fetal neurological injury and abnormalities [21,42]. Hamsters injected with OTA (2.5–20 mg/kg) during gestation (days 7–10) showed increased prenatal mortality, growth retardation and malformations such as micrognathia, hydrocephaly, oligodactyly, syndactyly, cleft lip, and heart defects [43]. In another in vivo study using a rat model, OTA (1–4 mg/kg) reduced maternal and embryonic weight, with higher doses causing complete embryo resorption [44]. Lower doses produced anomalies such as absence of tail, missing eyes, cleft lip, exencephaly, spina bifida, trunk curvature and limb reduction [44], thus supporting the present findings. An in vitro study demonstrated apoptosis and necrosis in human embryonic stem cells by OTA, highlighting its ability to induce cell death during early development [45].

FB_1_, a *Fusarium* toxin, is a hepatotoxic, nephrotoxic, and neurotoxic mycotoxin that was quantified in seven samples (18.4%) of serum in this study (Table 2), strongly indicating fetal exposure during pregnancy through maternal dietary intake, especially from contaminated cereals [1]. The available human data on FB_1_ exposure during developmental stages are based on indirect assessments of maternal biospecimens, typically serum or urine, which provide valuable indicators of maternal burden but do not fully capture placental transfer or fetal susceptibility. By contrast, the present findings obtained from stillborn fetal/neonatal samples offer more realistic evidence of in utero FB_1_ exposure. These results are supported by available data on maternal FB_1_ exposure. For example, in a cross-sectional study with 775 reproductive-age women in Guatemala, maternal urinary FB_1_ levels (up to 61.9 ng/mL) indicated that more than 75% of women exceeded the provisional maximum tolerable daily intake value, suggesting a higher in utero FB_1_ exposure [46]. However, despite low FB_1_ levels (nearly 1.0 ng/mL) being detected in over 93% of maternal biofluids in Rural Ethiopia, the overall exposure had no measurable adverse effect on birth outcomes [10]. These observations indicate that, while low-level exposure may not be associated with adverse outcomes, high dietary intake of FB_1_ can result in significant maternal and fetal health risks. FB_1_ can also induce cardiovascular disease in susceptible species, including humans, and has been implicated in neural tube defect (NTD) pathogenesis [47]. Reports have described its disruptive nature in the sphingolipid metabolism, impairing folate transport by interfering with folate receptor localization in lipid rafts [48,49]. A dose-dependent association between maternal FB_1_ exposure, assessed mainly via serum sphinganine-to-sphingosine ratios, indicated increased NTD incidence [49]. Mechanistic evidence from animal models further supports this relationship. Gelineau-van Waes et al. [50] demonstrated that oral administration of FB_1_ (20 mg/kg) to pregnant LM/Bc mice during early gestation resulted in a 79% incidence of NTDs. This exposure altered maternal and fetal sphingolipid profiles and significantly downregulated folate receptor (*Folbp1*) expression, reinforcing the role of FB_1_ as a teratogen through folate transport disruption. Additionally, FB_1_ enhances reactive oxygen species (ROS) production, resulting in oxidative stress, lipid peroxidation, and DNA damage. Apoptosis induced by FB_1_ may also stimulate compensatory proliferation, increasing the risk of malignant transformation [51]. Mechanistic insights showed that FB_1_ (0.313–5 μM) promoted proliferation, migration, and DNA damage in human esophageal epithelial cells. It also altered expression of cell cycle regulators and oncogenes, increased histone deacetylase activity, and activated the PI3K/Akt signaling pathway, which further supports FB_1_′s role in esophageal carcinogenesis [52]. FB_1_ reduces cardiac contractility and increases NTD risk in pregnancies. Impairment in the reproductive function, reducing gonadotropin levels, follicular development, and oocyte maturation are also reported due to FB_1_ [42]. In animal models, it is shown to induce embryonic defects, fetal malformations, and decreased litter sizes [53].

Further *Fusarium* toxins consist of estrogenic ZEN, which was quantified in one sample of liver tissue and in 13 (34.2%) serum samples evaluated in our work (Table 2). Studies indicated a rapid transfer of ZEN from the maternal to the fetal compartment, with significant amounts of the highly estrogenic phase—I metabolite α-ZEL and the less active ZEN-14-sulfate being released into both maternal and fetal circulation. Only limited studies have examined exposure to ZEN or its metabolites in humans across different life phases, including pregnant women [54,55,56]. In those studies, the prenatal exposure was assessed indirectly by assessing maternal samples or via the placenta or other biofluids, but none examined fetal/neonatal liver tissues or serum. However, ZEN ingestion through the contaminated foods during pregnancy can lead to fetal exposure not only to ZEN but also to its more potent α-ZEL (60-fold estrogenic than ZEN) [57]. In a pregnancy cohort from Connecticut (U.S.), ZEN was quantified as the key mycotoxin in 63% of maternal urine (median level: 0.16 ng/mL), with 2% of participants exceeding the established safety limits for ZEN [37]. Importantly, the more estrogenic metabolite α-ZEL was also detected as a key biomarker [37]. This is particularly concerning in pregnancy, as these estrogenic compounds may influence endocrine signaling and fetal development. Further cohort data provided clear evidence of endocrine disruption, with maternal myco-estrogen exposure linked to altered sex steroid hormone concentrations even at low levels [58]. In the UPSIDE cohort (*n* = 297), ZEN was detected in >93% of maternal urine samples, with median concentrations of 0.13 ng/mL (first trimester), 0.12 ng/mL (second trimester), and 0.32 ng/mL (third trimester) [55]. The authors also quantified α-ZEL in 76–90% of samples, reaching 0.22 ng/mL by the late pregnancy. In addition, the analysis of placentas (*n* = 118) showed the presence of ZEN (median 0.005 ng/g) and α-ZEL (up to 0.007 ng/g) in about 58% to 26% of samples, with total myco-estrogens (median 0.010 ng/g) quantified in 85% of samples. Notably, placental myco-estrogen burden predicted higher cord androgens (total and free testosterone) in male infants [58]. Another study in Rural Ethiopia confirmed the detection of ZEN and its metabolites (β-ZEL and β-zearalanol) in about half of the pregnant women, although no negative birth outcome was identified [10]. The negative effects posed by ZEN on the fetus are also supported by toxicological data from ex vivo or animal studies. ZEN exposure during early gestation disrupts placental development, impairs uterine and ovarian function, alters ATP-Binding Cassette (ABC) transporter expression, and compromises fetal growth and survival via hormone-sensitive vascular and molecular pathways [59]. Ex vivo perfusion study demonstrates that ZEN crosses the human placenta and is metabolized to its potent estrogenic derivatives that reach the fetal circulation [57]. High-dose animal studies show that maternal ZEN exposure leads to fetal growth restriction, exencephaly, exophthalmos and increased fetal loss [59], while peripubertal exposure disrupts fertilization and implantation [60]. ZEN’s endocrine-disrupting actions also reduce litter size and cause abnormal fetal growth and hormonal imbalances [61]. Thus, the detection of ZEN and its metabolites α-ZEL in one sample of liver and one of serum, as well as β-ZEL in one serum sample in the present study (Table 2), suggests a strong in utero or early life exposure to these metabolites. While β-ZEL is less potent and often present at low concentrations, α-ZEL exhibits strong estrogenic activity and may contribute to reproductive or developmental disturbances. These data support close monitoring of maternal ZEN exposure and its metabolites during pregnancy and highlight the need to assess potential synergism with other mycotoxins.

The occurrence levels of trichothecenes in liver tissue and serum samples analyzed in this work are corroborated by growing evidence from both adult human cohort studies and animal models that link dietary exposure to DON, T-2 and HT-2 toxins with adverse developmental and reproductive outcomes. Of these mycotoxins, DON occurs as a key contaminant in the majority of staple cereals and cereal-based foods [1,2]. This highlights the widespread and consistent exposure pattern among pregnant women worldwide. In a biomonitoring study conducted in Connecticut (U.S.), DON at a median level of 23 ng/mL was detected in 99% of urine from pregnant women, with 28–48% of the individuals surpassing the established maximum daily intake value for DON at different time points during pregnancy [37]. In a cohort of Caucasian pregnant women (*n* = 42) residing in East Yorkshire, United Kingdom, DON was detected in urine samples collected over two consecutive days [62]. Both free DON and DON-glucuronide were detected in 88.1% of women on day 1 and 83.3% on day 2 at a mean urinary level of 38.8 ng/mg creatinine (range: 0–269 ng/mg creatinine) for total DON. However, the authors concluded that the estimated ingestion of DON, calculated based on its urinary levels, was within the recommended tolerable daily intake for this mycotoxin [62]. Considering DON’s immunotoxicity and growth-suppressive effect, the high DON exposure raises significant maternal and fetal health concerns. A study from Rural Ethiopia also reported low DON levels in 39% of maternal urine samples, indicating an uncertain association with increased incidence of SGA, but no relationship between maternal DON exposure and adverse birth outcomes [10]. In contrast, data from a prospective human cohort study involving 1538 pregnant women in Wuhan, China, indicated that urinary DON levels correlated with decreased birth weight, also providing strong evidence of IUGR linked to maternal DON exposure [63]. Animal studies supported this evidence by demonstrating that maternal DON exposure during pregnancy and lactation adversely affects immune development, resulting in altered cytokine profiles and decreased regulatory T-cells, indicating notable immunotoxicity [64]. Further mechanistic data from in vitro and in vivo studies suggest broader reproductive risks by trichothecenes. High T-2 toxin doses induced DNA damage and chromosomal aberrations, also disrupting progesterone profiles and delaying ovulation [65]. Notably, in vitro data show that T-2 toxin crosses the placenta and accumulates in fetal tissues, where it may cause fetal brain injury and skeletal defects. Its metabolite, HT-2 toxin, impairs porcine oocyte maturation by disrupting spindle and actin dynamics, increasing ROS, and triggering apoptosis and autophagy [66], indicating compromised gamete quality and embryonic development. These mechanistic insights help contextualize our detection of DON, T-2 and HT-2 in liver and serum samples collected from the autopsied stillborn and neonatal patients. The human cohort evidence for DON clearly links maternal exposure to reduced birth weight and higher risk of small-for-gestational-age outcomes [63]. While direct human toxicological data regarding T-2 and HT-2 toxins remain limited, animal studies consistently show embryotoxicity, immunosuppression, and developmental defects following maternal exposure [65,66]. However, given the scarcity of human data, interpretations should remain cautious and based on careful extrapolation from animal evidence.

## 4. Conclusions

This study presents the first data on co-occurring mycotoxins in liver tissue and serum collected from autopsied stillborn and neonatal patients, offering compelling evidence of in utero mycotoxin exposure in Brazil. Elevated serum concentrations of OTA, FB_1_, and ZEN were found in some patients with structural malformations, neurodevelopmental and cardiovascular defects, growth impairments, and immunological vulnerabilities. Although potential associations between detectable mycotoxins in fetal biospecimens and congenital anomalies were indicated in this study, some limitations should be acknowledged. Firstly, the maternal dietary habits of patients evaluated, including food frequency questionnaires or food analysis, could not be verified during the trial, making it impossible to identify potential dietary sources of the mycotoxins evaluated. Secondly, only cases of autopsied patients were evaluated, leaving an open question regarding the mycotoxin levels that could be found in biospecimens from healthy patients. Thus, direct cause–effect relationships between the quantified mycotoxin levels in liver tissue or serum samples and the corresponding pathologies found in the evaluated patients remain to be determined. Importantly, the determination of correlations between the occurrence of congenital anomalies with maternal exposure to mycotoxins or other potential hazards was not included in the objectives of this work, as this would require a much different study design. Finally, the limited number of samples analyzed in the study reflects the mycotoxin exposure of the maternal group attended by CHRP in the state of São Paulo, but may not be representative of the exposure of the Brazilian general population to dietary mycotoxins. Given the scarcity of data on the mycotoxin intake by pregnant women in Brazil, investigating biomarkers in biospecimens collected from patients in other regions could contribute significantly to understanding the extent of dietary exposure during prenatal development at the national level.

Given the widespread occurrence of mycotoxins in food systems and their demonstrated embryotoxic potential, the findings of this study highlight an urgent need for proactive risk assessment, regulatory monitoring, and maternal dietary guidance during pregnancy. Integrating preventive measures to avoid fungi and mycotoxin contamination into maternal–child health initiatives is critical for reducing fetal and neonatal exposure to dietary mycotoxins during early stages of human development. Further studies should integrate data on dietary patterns of pregnant women, mycotoxin analyses of foods, and biomarkers of exposure and toxic effects, to provide robust evidence on the quantitative transplacental transfer of mycotoxins and their association with specific negative health effects during prenatal development.

## 5. Materials and Methods

### 5.1. Reagents and Solutions

All reagents used were of analytical grade, and solvents (acetonitrile and methanol) were HPLC-grade (JT Baker, Xalostoc, Mexico). Water was provided by a Milli-Q system (Millipore, Bedford, MA, USA). Mycotoxin standards (AFM_1_, AFB_1_, AFG_1_, AFG_2_, OTA, FB_1_, FB_2_, ZEN, α-ZEL, β- ZEL, DON, T-2 toxin and HT-2 toxin) were obtained from Sigma (St. Louis, MO, USA). Isotopic-labeled internal (IS) standards of [^13^C_17_]-AFM_1_, [^13^C_17_]-AFB_1_, [^13^C_20_]-OTA, [^13^C_34_]-FB_1_, [^13^C_18_]-ZEN, [^13^C_15_]-DON and [^13^C_24_]-T-2 toxin were purchased from Romer Labs (Getzersdorf, Austria). Aliquots of these individual IS were used to prepare a mixed solution in acetonitrile/water (1:1, *v/v*) containing [^13^C_17_]-AFM_1_, [^13^C_17_]-AFB_1_, [^13^C_20_]-OTA, [^13^C_34_]-FB_1_, [^13^C_18_]-ZEN and [^13^C_24_]-T-2 toxin at 20 ng/mL, and [^13^C_15_]-DON at 100 ng/mL.

### 5.2. Sampling Procedures

This study involved the collection of 43 liver tissues and 38 serum samples obtained from 23 neonates and 20 stillbirths during autopsies conducted between January 2019 and October 2024 at the Pathology Service of the Clinical Hospital of Ribeirão Preto (CHRP), Ribeirão Preto Medical School, University of São Paulo, Brazil. The mean and median postnatal ages of the neonates were 24 and 11 h, respectively. The stillbirths had mean and median gestational ages of 28.7 and 30 weeks, respectively. The study was submitted and approved by the Research Ethics Committee of CHRP (protocol no. 5.545.079/2022). Informed consent was obtained from the legal guardians of all cases.

A minimum of 3.0 g of liver tissue and 1.0 mL of whole blood per case were collected in cryovials. Blood samples were centrifuged at 3500 rpm for 10 min at 4 °C to separate serum, which, along with the liver tissues, was stored at −80 °C until analysis. All demographic and clinical details, including sex, medical record number, date and cause of death, body weight, congenital abnormalities, and maternal medical history, were retrieved from medical records. All cases were anonymized and identified solely by autopsy number, in compliance with Brazil’s General Data Protection Law [67].

### 5.3. Analysis of Mycotoxins in Serum and Liver Samples

Frozen liver tissues (*n* = 43) and serum samples (*n* = 38) were thawed at room temperature prior to analysis. Extraction procedures for liver samples followed the protocol by Cao et al. [68], with small modifications. Samples were finely ground in a cryogenic mill, and 1 g aliquots were weighed in 15-mL Falcon tubes containing 4 g of anhydrous sodium sulfate and 1.5 g of anhydrous sodium acetate. Next, 10 μL of the IS solution previously described and 10 mL of acetonitrile/water/acetic acid (79:20:1) were pipetted into the Falcon tubes, and the mixture was blended using an ultra-turrax homogeneizer (IKA T10 basic, Campinas, Brazil). Homogenized samples were centrifuged (5000× *g* for 8 min), and the supernatant was transferred to a clean Falcon tube. In total, 5 mL of *n*-hexane was added to the tube and left for 10 min in the dark. The *n*-hexane (upper) layer was discarded, and the acetonitrile phase was transferred to a glass tube for evaporation under nitrogen flux (45 ± 5 °C × 20 min). The resulting dried extracts were individually reconstituted in 500 μL of acetonitrile/water (10:90), vortexed (2 min), filtered (Nalgene^®^, 0.22 μm, Thermo Fisher Scientific Inc., Waltham, MA, USA), and transferred into 2 mL glass vials, which were stored at −20 °C until analysis.

For serum samples, the analytical protocol described by Cao et al. [68] was adapted with minor modifications. In brief, serum (800 μL) was incubated overnight at 37 °C with 10 μL of the previously prepared mixed IS solution and 40 μL of β-glucuronidase (2 mg/mL) in phosphate-buffered solution, for enzymatic deconjugation. Subsequently, 840 μL of acetonitrile and 84 μL of acetic acid were added, and the mixture was vortexed (5 min) and centrifuged at 7500× *g* for 5 min. The supernatant (750 μL) was loaded onto a MycoSpin™ 400 cleanup column (Romer Labs, Tulln, Austria) that underwent vortexing (1 min) and centrifugation (10,000 rpm × 1 min). Eluates were evaporated under nitrogen flux (45 ± 5 °C × 40 min) in a MultiVap54^®^ concentrator (LabTech, Sorisole, Italy), and the dried extracts were reconstituted in 340 μL of acetonitrile/water (10:90). Final extracts were filtered through a 0.22 μm Nalgene^®^ filter into 2 mL vials and stored at −20 °C until analysis.

### 5.4. Chromatographic and Mass Spectrometry Conditions

Final liver and serum extracts (5 μL each) were injected into a Waters Acquity I-Class^®^ ultraperformance liquid chromatography (UPLC) system (Waters, Milford, MA, USA), equipped with a BEH C_18_ column (2.1 × 50 mm, 1.7 μm) and coupled to a Xevo TQ-S^®^ triple quadrupole mass spectrometer (MS) (Waters, Milford, MA, USA). The column was maintained at 40 °C throughout the analysis, and the autosampler was kept at 15 °C. The mobile phase consisted of water (eluent A) and acetonitrile (eluent B), both containing 5 mM ammonium acetate and 0.1% acetic acid. The elution gradient began with 98% eluent A for the first 0.5 min, followed by a linear increase in eluent B to 30% over the next 4.5 min. Eluent B was then raised to 96% over a period of 2.0 min and maintained at this level for an additional 0.5 min, reaching a total run time of 7.5 min. Subsequently, eluent B was decreased to 2% over a 2.0 min period, and the system was re-equilibrated under initial conditions for another 2.0 min. The total chromatographic run time per sample was 10 min, with the mobile phase flow rate maintained at 0.6 mL/min.

Matrix-matched calibration curves were prepared using liver and serum extracts previously analyzed and found to have non-detectable levels of any mycotoxin under analysis. Six-point calibration curves were constructed with a mixture of mycotoxin standards dissolved in water/acetonitrile (9:1, *v*/*v*). For liver analysis, the concentrations were 0.10, 1.00, 5.00, 10.0, 50.0 and 100 ng/g for AFM_1_, AFB_1_, AFB_2_, AFG_1_, AFG_2_, OTA, FB_1_ and FB_2_; 1.00, 5.00, 10.0, 25.0, 50.0 and 100 ng/g for ZEN, α-ZEL, β- ZEL, T-2 toxin and HT-2 toxin; and 5.00, 10.0, 25.5, 50.0, 75.0 and 100 ng/mL for DON. For serum analysis, the concentrations of calibration curves were 0.06, 0.20, 0.50, 1.00, 2.00 and 4.00 ng/mL for AFM_1_, AFB_1_, AFB_2_, AFG_1_, AFG_2_, OTA, FB_1_ and FB_2_; 0.10, 0.25, 0.50, 1.00, 2.50 and 5.00 ng/mL for ZEN, α-ZEL, β- ZEL, T-2 toxin and HT-2 toxin; and 0.60, 1.25, 2.50, 5.00, 10.0 and 20.0 ng/mL for DON. Prior to injection, 20 µL of the mixed IS standard solution was added to 80 µL of the standard solution mixture in each calibration point.

MS was operated in Multiple Reaction Monitoring (MRM) mode using electrospray ionization (ESI) in both positive and negative ion modes. Capillary voltages were set to 3.00 kV for the positive mode and 2.00 kV for the negative mode. The source temperature was 150 °C, while the desolvation temperature was set at 500 °C. The desolvation gas flow was maintained at 800 L/h, and the cone gas flow at 150 L/h. The MRM quantification, confirmation transitions for mycotoxins, and optimized mass spectrometer parameters used for mycotoxin determinations in liver and serum samples are presented in Appendix A. All data acquisition and processing were performed using MassLynx software (v.4.1). The limits of detection (LOD) and quantification (LOQ) were defined using signal-to-noise ratios of 3 and 10, respectively (Appendix A). The analytical results were based on calibration curves generated in matrix-matched samples containing IS standards. These IS were used to normalize sample responses, which compensated for both recovery losses during sample processing and matrix effects [69].

### 5.5. Data Analysis

Concentrations of mycotoxin biomarkers in liver tissues and serum were plotted in Microsoft Excel^®^, and presented as descriptive results expressed as median and ranges (minimum–maximum). Liver and serum results from the same patients were applied to determine Pearson’s correlation coefficient between the levels of mycotoxin residues in both types of samples, using GraphPad Prism 7.0 (GraphPad, San Diego, CA, USA).

## Figures and Tables

**Figure 1 toxins-17-00518-f001:**
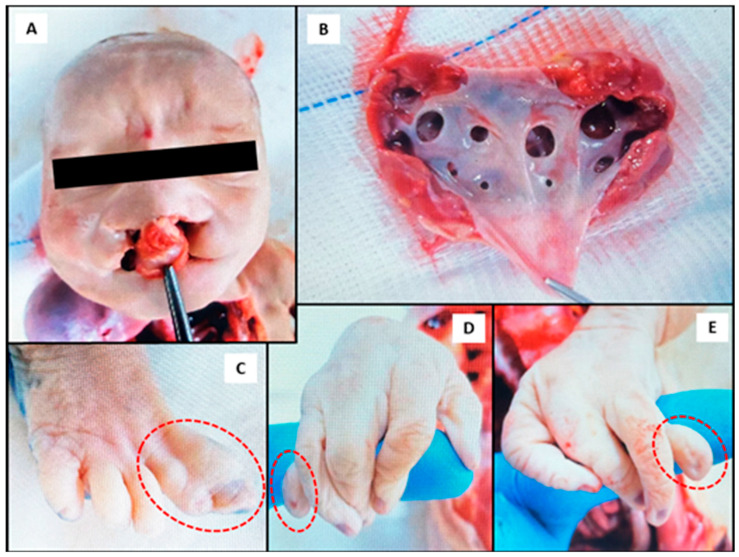
Photographs obtained during the autopsy of a neonate (case: 1082/2023) with multiple malformations and quantifiable levels of ochratoxin A (OTA) in the liver, as well as OTA, deoxynivalenol (DON) and T-2 toxin in serum. (**A**) Bilateral cleft lip and palate indicating craniofacial malformation; (**B**) bilateral hydronephrosis, more pronounced on the left kidney; (**C**) fusion of the first and second toes (indicated in the circle) on the right foot (syndactyly); (**D**,**E**) bilateral agenesis of the proximal and middle phalanges of the fifth fingers in the right (**D**) and left (**E**) hands (indicated in the circles), showing isolated distal phalanx development.

**Table 1 toxins-17-00518-t001:** Individual case data (demographics, clinical findings, causes of death) of neonates and stillborn fetuses autopsied in Ribeirão Preto, Brazil, and types of mycotoxins quantified in respective liver tissue and serum samples.

Case ID	Gender	Group	Age	Gestational Age	Birth Weight (kg)	Pathologies and/or Cause of Death	Mycotoxins in Liver ^1^	Mycotoxins in Serum ^2^
434/2019	F	Neonate	60 min	21 weeks + 3 d	0.400	Preterm premature rupture of membranes; Extreme prematurity	AFB_1_, AFB_2_, AFG_1_	OTA, FB_1_
455/2019	M	Stillborn	0	37 weeks	1.635	Intrauterine growth restriction; Hydrocephalus	-	ZEN
456/2019	M	Neonate	79 min	29 weeks + 4 d	2.155	Fetal hydrops	-	HT-2 toxin
481/2019	M	Neonate	120 min	21 weeks + 1 d	0.375	Intrauterine growth restriction; Extreme prematurity	-	OTA, DON
512/2019	M	Stillborn	0	35 weeks + 2 d	4.290	Imperfect twinning	-	AFM_1_
527/2019	F	Neonate	17.5 h	29 weeks + 2 d	1.140	Lung malformation	-	AFM_1_
64/2020	M	Stillborn	0	26 weeks + 2 d	0.570	Preterm premature rupture of membranes; Twin pregnancy	-	-
66/2020	M	Neonate	41 min	20 weeks + 2 d	0.320	Extreme prematurity	AFM_1_	DON
69/2020	F	Neonate	16.5 h	22 weeks	0.485	Extreme prematurity	-	AFM_1_, ZEN, α-ZEL
73/2020	F	Neonate	17 h	38 weeks + 3 d	1.715	Turner syndrome; Neonatal anoxia	-	AFM_1_, ZEN
76/2020	M	Stillborn	0	30 weeks	1.572	Maternal death	-	AFM_1_, FB_1_, ZEN, β-ZEL, HT-2 toxin
101/2020	F	Neonate	126 min	22 weeks + 4 d	0.520	Preterm premature rupture of membranes; Acute chorioamnionitis; Extreme prematurity	-	AFM_1_, OTA, FB_1_
484/2022	F	Neonate	8.5 h	37 weeks	ND	Placental abruption	-	-
732/2022	M	Stillborn	0	37 weeks + 6 d	2.550	Preeclampsia	AFM_1_	OTA, FB_1_
83/2023	F	Stillborn	0	21 weeks + 5 d	0.254	Intrauterine growth restriction	OTA	AFM_1_
158/2023	F	Stillborn	0	20 weeks	1.325	Preterm premature rupture of membranes	-	-
170/2023	F	Neonate	15.5 h	25 weeks	0.730	Extreme prematurity	OTA	AFM_1_, ZEN
237/2023	M	Stillborn	0	34 weeks + 4 d	ND	Intrauterine anoxia	-	AFM_1_, ZEN
307/2023	F	Neonate	14.3 h	22 weeks + 6 d	0.460	Twin pregnancy; Extreme prematurity	-	OTA, ZEN
363/2023	F	Stillborn	0	22 weeks + 4 d	0.305	Intrauterine growth restriction	-	AFM_1_, OTA
461/2023	M	Neonate	6 d	33 weeks + 2 d	1.090	Edwards syndrome	-	AFM_1_, ZEN
464/2023	M	Stillborn	0	ND	ND	Absolute oligoamnios; Potter sequence	-	AFM_1_
555/2023	F	Stillborn	0	26 weeks	0.960	Intrauterine anoxia	-	OTA
673/2023	M	Stillborn	0	35 weeks	ND	Intrauterine growth restriction; Congenital syphilis	ZEN	Na
766/2023	M	Stillborn	0	30 weeks + 1 d	1.005	Preterm premature rupture of membranes	-	AFM_1_, DON
796/2023	M	Neonate	24 min	28 weeks + 5 d	1.085	Intrauterine growth restriction; Twin pregnancy	-	ZEN
797/2023	M	Stillborn	0	28 weeks + 5 d	0.795	Twin-to-twin transfusion syndrome	-	ZEN
809/2023	M	Stillborn	0	21 weeks + 5 d	0.480	ND	-	OTA, FB_1_,
813/2023	M	Neonate	6 d	35 weeks	3.550	Maternal sepsis	α-ZEL	OTA, DON
816/2023	M	Neonate	61 min	38 weeks + 1 d	2.340	Multicystic dysplastic kidney; Potter sequence	-	-
870/2023	F	Stillborn	0	24 weeks + 5 d	0.490	Turner syndrome; congenital heart malformation	-	Na
918/2023	M	Neonate	70 min	32 weeks + 5 d	1.800	Potter sequence	-	Na
975/2023	F	Stillborn	0	31 weeks	1.120	Maternal arterial hypertension	OTA	ZEN
978/2023	F	Stillborn	0	20 weeks	0.230	Twin-to-twin transfusion syndrome	-	DON
1027/2023	F	Neonate	99 min	22 weeks + 5 d	0.434	Preterm premature rupture of membranes; Extreme prematurity	-	-
1082/2023	M	Neonate	9.3 h	35 weeks + 1 d	1.680	Multiple congenital malformations	OTA	OTA, DON, HT-2 toxin
1090/2023	F	Neonate	19 h	35 weeks	2.280	Congenital cardiac and intestinal malformations	-	DON
1111/2023	F	Stillborn	0	23 weeks	0.380	Preterm premature rupture of membranes	-	Na
13/2024	F	Neonate	3.5 d	27 weeks	0.960	HELLP syndrome; Neonatal sepsis	-	OTA
75/2024	F	Neonate	23 h	39 weeks + 4 d	2.990	Congenital heart malformation	-	AFM_1_, OTA, DON
84/2024	F	Neonate	13 h	30 weeks	1.520	Preterm premature rupture of membranes; Prematurity	-	AFM_1_, OTA, ZEN, DON
206/2024	F	Neonate	60 min	21 weeks	ND	Congenital heart malformation	-	FB_1_, T-2, HT-2 toxins
290/2024	F	Stillborn	0	32 weeks	1.280	Maternal arterial hypertension	-	Na

^1^ Liver tissue samples (*n* = 9) containing levels ≥ the limit of quantification (LOQ) of mycotoxins (see Appendix A for LOQ values of each mycotoxin). ^2^ Serum samples (*n* = 33) containing levels ≥ the LOQ of mycotoxins (see Appendix A for LOQ values of each mycotoxin). F: female; M: male; ND: not determined; HELLP: Hemolysis, Elevated Liver enzymes, and Low Platelet count; Na: sample not available at the time of liver collection. AFB_1_: aflatoxin B_1_; AFB_2_: aflatoxin B_2_; AFG_1_: aflatoxin G_1_; AFM_1_: aflatoxin M_1_; OTA: ochratoxin A; FB_1_: fumonisin B_1_; ZEN: zearalenone; α-ZEL: α-zearalenol; β-ZEL: β-zearalenol; DON: deoxynivalenol.

**Table 2 toxins-17-00518-t002:** Mycotoxins quantified in liver tissues and serum samples from stillborn fetuses and neonates in Ribeirão Preto, Brazil.

Mycotoxin	Liver Tissue Samples (*n* = 43)			Serum Samples (*n* = 38)		
	*n* ^1^ (%)	Median (ng/g)	Range (ng/g)	*n* ^1^ (%)	Median (ng/mL)	Range (ng/mL)
AFM_1_	2 (4.7)	0.38	0.34–0.42	15 (39.5)	0.48	0.20–4.30
AFB_1_	1 (2.3)	1.79	1.79	0	<LOQ	-
AFB_2_	1 (2.3)	0.30	0.30	0	<LOQ	-
AFG_1_	1 (2.3)	0.62	0.62	0	<LOQ	-
AFG_2_	0	<LOQ	-	0	<LOQ	-
OTA	4 (9.3)	0.31	0.30–0.34	13 (34.2)	3.39	1.11–157.8
FB_1_	0	<LOQ	-	7 (18.4)	30.6	0.90–218.1
FB_2_	0	<LOQ	-	0	<LOQ	-
ZEN	1 (2.3)	3.60	3.60	13 (34.2)	10.53	1.32–198.9
α-ZEL	1 (2.3)	2.76	2.76	1 (2.6)	4.15	4.15
β-ZEL	0	<LOQ	-	1 (2.6)	18.41	18.41
DON	0	<LOQ	-	9 (23.7)	5.71	3.27–23.94
T-2 toxin	0	<LOQ	-	1 (2.6)	2.85	2.85
HT-2 toxin	0	<LOQ	-	4 (10.5)	10.84	5.92–62.04

^1^ Number of samples containing levels ≥ the limit of quantification (LOQ) of mycotoxins (see Appendix A for LOQ values of each mycotoxin). AFM_1_: aflatoxin M_1_; AFB_1_: aflatoxin B_1_; AFB_2_: aflatoxin B_2_; AFG_1_: aflatoxin G_1_; AFG_2_: aflatoxin G_2_; OTA: ochratoxin A; FB_1_: fumonisin B_1_; FB_2_: fumonisin B_2_; ZEN: zearalenone; α-ZEL: α-zearalenol; β-ZEL: β-zearalenol; DON: deoxynivalenol.

## Data Availability

The original contributions presented in this study are included in the article/Appendix A. Further inquiries can be directed to the corresponding authors.

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
