# Peer review of "Assessment of Maternal Exposure to Mycotoxins During Pregnancy Through Biomarkers in Fetal and Neonatal Tissues"

_toxins, 2025, doi:10.3390/toxins17100518_

Round 1

Reviewer 1 Report

Comments and Suggestions for Authors

In this study, the authors have assessed maternal exposure to mycotoxins during pregnancy in fetal and neonatal tissues. Although the data presented is interesting, the authors need to clarify the following concerns:

  1. As the authors identified themselves as a limitation, this is more observational study without establishing any correlation between congenital anomalies and mycotoxins. There is already such research articles published earlier, questioning the novelty of this study?
  2. Congenital anomalies, in addition to the exposure to toxin, can be a result of genetic factors, drugs or even infection. In the present study, authors do not present any such possibilities like the medications taken by the pregnant women and/or whether they had any infection during the period of gestation.
  3. Did the authors record the mycotoxin levels in maternal serum samples, or detect them in their urine samples?
  4. Line 9: Is ‘patient’ a correct word here?
  5. Line 9: What does AFM1 stand for (it is appearing for the first time, though it has been expanded later on in the text)?
  6. Lines 298-300: what is the connection between immunotoxicity, mycotoxins and congenital anomalies? Could the authors describe it better?
  7. Sample size looks to be limited compared to previous studies.
  8. I could not see ethical details presented in the manuscript; was it not necessary though the samples were from still births/autopsy?
  9. Protocols need to be explained better; for e.g., liver extract was finally reconstituted in aceto-nitrile-water, and the serum extracts were reconstituted in methanol. Was there any specific reason?

Author Response

Reviewer #1: In this study, the authors have assessed maternal exposure to mycotoxins during pregnancy in fetal and neonatal tissues. Although the data presented is interesting, the authors need to clarify the following concerns:

Answer: We thank the reviewer for their important comments. We believe that all concerns have now been addressed in the revised version of the manuscript.

  1. As the authors identified themselves as a limitation, this is more observational study without establishing any correlation between congenital anomalies and mycotoxins. There is already such research articles published earlier, questioning the novelty of this study?

Answer: Yes, several studies have described associations between maternal exposure to some mycotoxins and adverse birth outcomes, often through indirect assessments of biomarkers in maternal urine or serum. However, the objective of the present study was to evaluate the maternal exposure to mycotoxins during pregnancy through biomarkers in biospecimens from stillbirths and neonates, not to correlate the occurrence of congenital anomalies with fetal exposure to mycotoxins. To our knowledge, this is the first study describing multiple mycotoxin residues directly measured in stillborn/neonatal liver and serum samples, thus providing direct confirmation that mycotoxins can cross the human placental barrier and accumulate in fetal tissues as well as enter the prenatal circulation. Also, this is the first report on maternal exposure to dietary mycotoxins during pregnancy in Brazil, which corroborate for the novelty of the study. These aspects were amended in the Introduction section of the revised manuscript.

  1. Congenital anomalies, in addition to the exposure to toxin, can be a result of genetic factors, drugs or even infection. In the present study, authors do not present any such possibilities like the medications taken by the pregnant women and/or whether they had any infection during the period of gestation.

Answer: Thanks for the comment. The authors agree that additional information on the genetic factors, drugs or even infection could shed light on the possible causes of death of the evaluated cases. But, again, the determination of correlations between the occurrence of congenital anomalies with maternal exposure to mycotoxins or other potential hazards was not included in the objectives of the work, as this would require a much different study design. Given the limited number of cases evaluated, only specific outcomes regarding high mycotoxin levels found in liver and/or serum could be addressed in the discussion (e.g., patient no. 1082/2023).

  1. Did the authors record the mycotoxin levels in maternal serum samples, or detect them in their urine samples?

Answer: Unfortunately, because of ethical constraints, it was not possible to assess maternal serum or urine sample at the time of the study.

  1. Line 9: Is ‘patient’ a correct word here?

Answer: Yes, it is correct. The term ‘patient’ comes from the Latin word “patiens”, meaning ‘one who suffers’, referring to a person who requires care from a healthcare professional (a doctor, nurse, or other specialist). Even if the patient has already died, an autopsy is the medical procedure aimed at clarifying relevant aspects related to his/her illness and death.

  1. Line 9: What does AFM1 stand for (it is appearing for the first time, though it has been expanded later on in the text)?

Answer: Done.

  1. Lines 298-300: what is the connection between immunotoxicity, mycotoxins and congenital anomalies? Could the authors describe it better?

Answer: The connection between immunotoxicity, mycotoxins and congenital anomalies were amended in the Discussion section of the revised manuscript.

  1. Sample size looks to be limited compared to previous studies.

Answer: As described in section 5.2 of the original manuscript, the number of liver and serum samples evaluated reflects the available number of autopsies conducted in neonates and stillbirths between January 2019 and October 2024.

  1. I could not see ethical details presented in the manuscript; was it not necessary though the samples were from still births/autopsy?

Answer: Yes, of course an ethical statement was necessary, and this was declared in the original version of the manuscript (please see L.382-385 of section 5.2). The Institutional Review Board Statement was approved by the ethics committee on 26th of July 2022.

  1. Protocols need to be explained better; for e.g., liver extract was finally reconstituted in aceto-nitrile-water, and the serum extracts were reconstituted in methanol. Was there any specific reason?

Answer: Thank you for pointing out this issue. In fact, the reconstitution solvent was wrongly written in the original manuscript as both matrices were reconstituted in the same mixture (acetonitrile:water, 10:90). This was corrected in the revised manuscript.

Reviewer 2 Report

Comments and Suggestions for Authors

The present study aimed at investigating the transplacental transfer of mycotoxins during human prenatal development during fetal and neonatal tissues. The study was interesting and important as this is the first study indicating maternal exposure to dietary mycotoxins during pregnancy in Brazil. The paper was well-designed and written in good English. The current manucript could be published in Toxins after only minor revisions as follows.

  1. Since neonatal tissues were used, the ethical statment and approved code should be provided.
  2. In the introduction, the authors should describe the mycotoxins limit in food set by the local government.
  3. There are many abbreviations in the manuscript. The authors should add an abbreviation list to help reading.
  4. The conclusion should be combined in one part and the authors should stated the limitation of this study as the sample number was not enough to draw concluion by statistial analysis.

Author Response

Reviewer #2: The present study aimed at investigating the transplacental transfer of mycotoxins during human prenatal development during fetal and neonatal tissues. The study was interesting and important as this is the first study indicating maternal exposure to dietary mycotoxins during pregnancy in Brazil. The paper was well-designed and written in good English. The current manucript could be published in Toxins after only minor revisions as follows.

Answer: Thanks for summarizing the main findings of the study.

  1. Since neonatal tissues were used, the ethical statment and approved code should be provided.

Answer: The ethical statement was declared in the original version of the manuscript (please see L.382-385 of section 5.2). The Institutional Review Board Statement was approved by the ethics committee on 26th of July 2022.

  1. In the introduction, the authors should describe the mycotoxins limit in food set by the local government.

Answer: The maximum limits adopted in Brazil for regulated mycotoxins were amended in the Introduction section, as requested.

  1. There are many abbreviations in the manuscript. The authors should add an abbreviation list to help reading.

Answer: Thanks for the suggestion. However, it was not possible to find a proper section in the manuscript template provided by the journal to add the abbreviation list. Nevertheless, all abbreviations mentioned throughout the text were double checked for their respective explanation before their first use in the manuscript.

  1. The conclusion should be combined in one part and the authors should stated the limitation of this study as the sample number was not enough to draw concluion by statistial analysis.

Answer: Done.

Reviewer 3 Report

Comments and Suggestions for Authors

Review

Manuscript entitled „Assessment of Maternal Exposure to Mycotoxins During Preg-2 nancy Through Biomarkers in Fetal and Neonatal Tissues” presents very interesting results on a very important topic, but there are many shortcomings that need to be corrected. The article cannot be published in present form.

My main objection concerns the design of the study. The study only looked at dead fetuses and did not give consideration to healthy newborns. It is already known that mycotoxins cross the placenta, but it is not known whether they cause birth defects. Therefore, it would be worthwhile to test the blood serum of healthy children. Such studies would answer many questions about the impact of mycotoxins on fetal development. After all, there may be other correlations that will not be explained until newborns who were born healthy but were exposed to mycotoxins are given consideration.

line 34, 38 and 41 - literature reference 1-3—the literature should refer to Fusarium fungi and their metabolites and should be more mycological in nature.

Additionally, I believe that research articles are more valuable than reviews. Therefore, I would suggest replacing the cited reviews with specific articles (references: 3, 12,13,14, 17)

line 46 and 52 - Studies on mycotoxins should be cited, rather than those on pregnancy, newborns, or methods for their detection.

line 88 and line -183-186 - There are many studies on the occurrence of mycotoxins in body fluids. Throughout the entire study, there is no reference to these articles, for example:

Shuaib, F.M.B.; Jolly, P.E.; Ehiri, J.E.; Yatich, N.; Jiang, Y.; Funkhouser, E.; Person, S.D.; Wilson, C.; Ellis, W.O.; Wang, J.-S.; et al. Association between birth outcomes and aflatoxin B1biomarker blood levels in pregnant women in Kumasi, Ghana. Trop. Med. Int. Health 2010, 15, 160–167. 

Abulu, E.O.; Uriah, N.; Aigbefo, H.S.; Oboh, P.A.; Agbonlahor, D.E. Preliminary investigation on aflatoxin in cord blood of jaundiced neonates. West Afr. J. Med. 1998, 17, 184–187.

Malir, F.; Ostry, V.; Dofkova, M.; Roubal, T.; Dvorak, V.; Dohnal, V. Ochratoxin A levels in blood serum of Czech women in the first trimester of pregnancy and its correspondence with dietary intake of the mycotoxin contaminant. Biomarkers 2013, 18, 673–678.

Wells, L.; Hardie, L.; Williams, C.; White, K.; Liu, Y.; De Santis, B.; Debegnach, F.; Moretti, G.; Greetham, S.; Brera, C.; et al. Determination of Deoxynivalenol in the Urine of Pregnant Women in the UK. Toxins 2016, 8, 306. 

Gromadzka, K., Pankiewicz, J., Beszterda, M., Paczkowska, M., Nowakowska, B., & KocyÅ‚owski, R. (2021). The Presence of Mycotoxins in Human Amniotic Fluid. Toxins, 13(6), 409. 

line 175-177 - Why were the correlation results not included? I believe it would be worthwhile to check all possible correlations, e.g., between toxin content and gender, between toxin content and birth weight, between toxin type and birth defects, etc.

line 197-200 - It is not known whether mycotoxins have an impact on birth defects, as the levels of these compounds in healthy newborns have not been determined. In addition, there have been cases where congenital defects were present but no mycotoxins were found.

line 264-266 - no reference

In the section on the discussion of results, I would add studies on the content of mycotoxins in body fluids in pregnant women and newborns. I would not focus just on animal studies.

Author Response

Reviewer #3: Manuscript entitled „Assessment of Maternal Exposure to Mycotoxins During Preg-2 nancy Through Biomarkers in Fetal and Neonatal Tissues” presents very interesting results on a very important topic, but there are many shortcomings that need to be corrected. The article cannot be published in present form.

Answer: Thanks for the detailed review. The manuscript was revised according to your comments and suggestions.

My main objection concerns the design of the study. The study only looked at dead fetuses and did not give consideration to healthy newborns. It is already known that mycotoxins cross the placenta, but it is not known whether they cause birth defects. Therefore, it would be worthwhile to test the blood serum of healthy children. Such studies would answer many questions about the impact of mycotoxins on fetal development. After all, there may be other correlations that will not be explained until newborns who were born healthy but were exposed to mycotoxins are given consideration.

Answer: The authors agree that additional data from healthy newborns would provide new insights on the impact of mycotoxins on fetal development. However, it was not possible to obtain liver or blood samples from healthy children, as all cases in this study involved autopsied patients whose causes of death were associated with congenital malformations and/or severe clinical conditions capable of leading to death. Importantly, the objective of the present study was to evaluate the maternal exposure to mycotoxins during pregnancy through biomarkers in biospecimens from stillbirths and neonates, not to correlate the occurrence of congenital anomalies with fetal exposure to mycotoxins because this would require a much different study design.

line 34, 38 and 41 - literature reference 1-3—the literature should refer to Fusarium fungi and their metabolites and should be more mycological in nature.

Answer: The literature references were updated in those lines along with clarifications regarding the fungi species associated with each type of mycotoxin, as requested.

Additionally, I believe that research articles are more valuable than reviews. Therefore, I would suggest replacing the cited reviews with specific articles (references: 3, 12,13,14, 17)

Answer: These references (reviews) were updated with research-based evidences, as suggested.

line 46 and 52 - Studies on mycotoxins should be cited, rather than those on pregnancy, newborns, or methods for their detection.

Answer: It was not possible to address this comment, as those lines in the original version of the manuscript deal with the carryover of mycotoxins into animal-derived foods and maximum levels for mycotoxins in foods adopted in Brazil.

line 88 and line -183-186 - There are many studies on the occurrence of mycotoxins in body fluids. Throughout the entire study, there is no reference to these articles, for example:

Shuaib, F.M.B.; Jolly, P.E.; Ehiri, J.E.; Yatich, N.; Jiang, Y.; Funkhouser, E.; Person, S.D.; Wilson, C.; Ellis, W.O.; Wang, J.-S.; et al. Association between birth outcomes and aflatoxin B1biomarker blood levels in pregnant women in Kumasi, Ghana. Trop. Med. Int. Health 2010, 15, 160–167.

Abulu, E.O.; Uriah, N.; Aigbefo, H.S.; Oboh, P.A.; Agbonlahor, D.E. Preliminary investigation on aflatoxin in cord blood of jaundiced neonates. West Afr. J. Med. 1998, 17, 184–187.

Malir, F.; Ostry, V.; Dofkova, M.; Roubal, T.; Dvorak, V.; Dohnal, V. Ochratoxin A levels in blood serum of Czech women in the first trimester of pregnancy and its correspondence with dietary intake of the mycotoxin contaminant. Biomarkers 2013, 18, 673–678.

Wells, L.; Hardie, L.; Williams, C.; White, K.; Liu, Y.; De Santis, B.; Debegnach, F.; Moretti, G.; Greetham, S.; Brera, C.; et al. Determination of Deoxynivalenol in the Urine of Pregnant Women in the UK. Toxins 2016, 8, 306.

Gromadzka, K., Pankiewicz, J., Beszterda, M., Paczkowska, M., Nowakowska, B., & Kocyłowski, R. (2021). The Presence of Mycotoxins in Human Amniotic Fluid. Toxins, 13(6), 409.

Answer: Thanks for your suggestion. All of the above articles were included in the discussion of the revised manuscript.

line 175-177 - Why were the correlation results not included? I believe it would be worthwhile to check all possible correlations, e.g., between toxin content and gender, between toxin content and birth weight, between toxin type and birth defects, etc.

Answer: The correlation data between the levels of mycotoxins in the liver and serum samples was amended in the revised Supplementary material, as Figure S1. Other potential correlations (e.g., between toxin content and gender, between toxin content and birth weight, between toxin type and birth defects), although very interesting, unfortunately were not possible to assess due to the limited number of units (samples) in each category (e.g., liver or serum samples containing each type of mycotoxin among genders or birth weights or birth defects).

line 197-200 - It is not known whether mycotoxins have an impact on birth defects, as the levels of these compounds in healthy newborns have not been determined. In addition, there have been cases where congenital defects were present but no mycotoxins were found.

Answer: The authors agree that additional data from healthy newborns would provide new insights on the impact of mycotoxins on fetal development. However, it was not possible to obtain liver or blood samples from healthy children, as all cases in this study involved autopsied patients whose causes of death were associated with congenital malformations and/or severe clinical conditions capable of leading to death. Importantly, the objective of the present study was to evaluate the maternal exposure to mycotoxins during pregnancy through biomarkers in biospecimens from stillbirths and neonates, not to correlate the occurrence of congenital anomalies with fetal exposure to mycotoxins because this would require a much different study design.

line 264-266 - no reference

Answer: A proper reference was added to support the statement.

In the section on the discussion of results, I would add studies on the content of mycotoxins in body fluids in pregnant women and newborns. I would not focus just on animal studies.

Answer: The Discussion section of the revised manuscript was amended with studies describing the levels of mycotoxins found in body fluids of pregnant women and certain related matrices such as placenta or cord blood. To our knowledge, there is no direct experimental or interventional evidence describing mycotoxin residues in human fetuses or newborns. Human observational cohorts primarily report associations (e.g., low birth weight, growth restriction), but not congenital anomalies in a causal or controlled sense. Furthermore, human biomonitoring/observational studies rarely include detailed congenital malformation data directly associated with toxin levels.

Round 2

Reviewer 1 Report

Comments and Suggestions for Authors

The authors have answered the queries raised.

Author Response

Thank you!

Reviewer 3 Report

Comments and Suggestions for Authors

 The authors responded to my review in detail and gave consideration to my suggestions in the revised manuscript. My only comment concerns the authors' response that “the aim of this study was to assess maternal exposure to mycotoxins during pregnancy through biomarkers in biological samples from stillborn infants and newborns, rather than to correlate the occurrence of birth defects with fetal exposure to mycotoxins, as this would require a completely different study design.” If this were indeed the aim of the study, it would suffice to analyze the blood of pregnant women, rather than the liver and blood of dead fetuses/newborns. In addition, the manuscript repeatedly states that the presence of mycotoxins potentially affects placental function, fetal development, and postnatal viability. Such statements appear in the discussion and conclusions. In my opinion, without blood tests of healthy newborns, such a statement is unacceptable. I leave the decision to accept the manuscript for publication to the editor.

Author Response

The authors thank the additional comments and suggestions of the Reviewer #3 for improving the manuscript. We have addressed all of them and included below a point-by-point response to the Reviewer. The changes in the text were done only by exclusion of some sentences in the Discussion and Conclusion sections of the revised manuscript, as suggested by the Reviewer.

Reviewer #3: The authors responded to my review in detail and gave consideration to my suggestions in the revised manuscript. My only comment concerns the authors' response that “the aim of this study was to assess maternal exposure to mycotoxins during pregnancy through biomarkers in biological samples from stillborn infants and newborns, rather than to correlate the occurrence of birth defects with fetal exposure to mycotoxins, as this would require a completely different study design.” If this were indeed the aim of the study, it would suffice to analyze the blood of pregnant women, rather than the liver and blood of dead fetuses/newborns. In addition, the manuscript repeatedly states that the presence of mycotoxins potentially affects placental function, fetal development, and postnatal viability. Such statements appear in the discussion and conclusions. In my opinion, without blood tests of healthy newborns, such a statement is unacceptable. I leave the decision to accept the manuscript for publication to the editor.

Answer: The authors agree with the comments on the unacceptable statements, so they were excluded from the Discussion and Conclusion sections of the revised manuscript. But the authors respectfully disagree with the comment “If this were indeed the aim of the study, it would suffice to analyze the blood of pregnant women, rather than the liver and blood of dead fetuses/newborns”. Analysis of blood from pregnant women is an indirect approach to assess the mycotoxin exposure of fetuses/newborns. However, several factors may affect the rates of transplacental crossing by different mycotoxins, including the duration and gestational time of exposure, maternal metabolism and health status, and the presence of conjugated metabolites of mycotoxins (Woo et al. 2012, Toxicol. Lett. 2012, 208, 92-99, doi:10.1016/j.toxlet.2011.10.013). Therefore, the levels of mycotoxins in the maternal blood do not indicate the effective internal dose of mycotoxins that reaches the fetuses/newborns. For these reasons, the data presented in the manuscript indicate for the first-time direct, quantitative evidence of human prenatal in-utero exposure to dietary mycotoxins.